# Human–Robot Collaborative Assembly Based on Eye-Hand and a Finite State Machine in a Virtual Environment

**Xue Zhao, Ye He \*, Xiaoan Chen and Zhi Liu**

State Key Laboratory of Mechanical Transmission, Chongqing University, Chongqing 400044, China; 20170701003@cqu.edu.cn (X.Z.); xachen@cqu.edu.cn (X.C.); 20112914@cqu.edu.cn (Z.L.)
\* Correspondence: h1166@cqu.edu.cn

**Abstract:** With the development of the global economy, the demand for manufacturing is increasing. Accordingly, human–robot collaborative assembly has become a research hotspot. This paper aims to solve the efficiency problems inherent in traditional human-machine collaboration. Based on eye–hand and finite state machines, a collaborative assembly method is proposed. The method determines the human's intention by collecting posture and eye data, which can control a robot to grasp an object, move it, and perform co-assembly. The robot's automatic path planning is based on a probabilistic roadmap planner. Virtual reality tests show that the proposed method is more efficient than traditional methods.

**Keywords:** human–robot interaction; eye-tracking; gesture recognition; virtual reality; assembly

## 1. Introduction

Global competition in manufacturing is becoming increasingly fierce, with greater consumer demand for high-quality but less expensive products [1]. Much labor and materials are used in the assembly process and for public facilities and maintenance [2]. The cumbersome nature of manual assembly creates inefficiency issues for workers [3]. However, full automation of assembly lines is currently too expensive and inflexible [4]. Combining the advantages of automation (reliability and stability) with those of people (flexibility and adaptability) can make assembly processes more flexible, cheap, and productive [5].

Human–robot collaboration (HRC) is where humans and machines work together to accomplish tasks. The humans are responsible for controlling and monitoring production, while the robots do hard physical work [6]. The excellent characteristics of each can be combined, which is the core concept of "industry 4.0." Petruck [7] proposed an alternative configuration of HRC workplaces called "CoWorkAs," which combines human cognitive and sensorimotor skills with the precision, speed, and fatigue-free operation of robots to achieve effective collaboration. HRC has become a research hotspot and is widely used, including in elderly care, space stations, rescue, and assembly [8,9]. A review of state-of-the-art physical human–robot collaborative assembly tasks was introduced in [10,11]. HRC can solve various industry problems and so has become an area worthy of study [12]. Research in this area studies the responses of the human body to the adaptive level of robots to determine the impacts on team fluency, human satisfaction, safety, and comfort. Magrini [13] studied the safety of human–robot interactions and used gestures to change the operating modes of robots.

Previous interaction methods have indeed met the functional requirements; however, the interaction is inefficient. With the advent of advanced virtual reality (VR) assembly technology, physical assemblies can be presented in a virtual environment [14]. There are various interaction models, such as speech, electroencephalography, gestures, and eye movement [15,16]. Among them, gesture recognition has proved to be the most natural and immersive way for users to interact with a 3D environment [17]. Gesture recognition technology has been reported to increase machine manufacturing efficiency [18]. Kim [19]

used a depth camera to collect human posture data and minimize the effect of external loads on body joints. Teleoperation technology with visual feedback has been used in aerospace mask teaching, enabling robots to learn skillful and complex manipulation tasks [20]. The above interactive technology has mainly used human posture data. Zhi [21] found that face direction and eye gaze data with gestures help improve pointing estimation. Mayer [22] found that the accuracy of eye–finger combinations is better than that of a finger, forearm, or head radiation.

Among these methods, voice communication is commonly used in human interaction. However, it can be hard for robots to apply. For example, when you want the robot to move to the left, it is hard for the robot to understand; the robot needs additional parameters, such as distance, acceleration, and directions relative to the environment or itself. It is even more challenging when it comes to rotation and run curves. Then, we can define forward, backward, rotation, and other commands through gestures. However, there few gestures commonly used by humans, so the number of these gestures is not enough, and combinations of multiple gestures increase the burden of interaction. An interaction model based on hand and arm postures is flexible, but the user still needs to complete whole actions, which creates a burden. In addition, methods based on electroencephalography have great potential but are still immature and, due to their stability and complexity, research is still required. Lastly, eye movement (which mainly provides attention to things) could be used to control a mouse cursor to perform simple tasks on a computer. However, it is difficult to control multi-degree objects in this way.

Each of the operating environments mentioned above is very different, yet the core principle is the same: master–slave mode. The human acts as a master terminal providing operation instructions, while the robot performs specific tasks as the slave. Currently, research is focusing on intelligent learning from simulations that exploit AI algorithms [23,24]; improving the interaction efficiency and reducing the work intensity and the cost are the goals. For example, some studies have explored the learning and modeling of human-operated regions by locally weighted regression [25] or Gaussian mixture regression [26]. The state region is where the established local model is controlled by a robot; otherwise, it is controlled by humans. Additionally, some slight indicative actions can trigger a robot to complete certain tasks via eye contact and brain signals [27], which reduces the human's work effort.

In this study, indicative actions and mapping gestures were used to reduce the effort of user engagement. The HRC model combines object recognition, gesture recognition, and a finite state machine (FSM). Semiautomatic operations, including grasping and placing modes, were devised, and object verification was done in a virtual environment. The robot identifies the object and plans its trajectory and movement by a probabilistic roadmap planner (PRM) [28] algorithm. The user only provides some demand hints so that the robot can grab the appropriate objects. On the other hand, the robot can be seamlessly switched to a mapping mode to place an object, which provides high flexibility.

The highlights of this study are:

(1) The robot and scene are built in a Unity environment, and the robot joints are controlled by the kinematics model in Simulink software.
(2) Hand data is collected by a Leap-motion sensor. We divide hand gestures into nine categories, use the joint angles of the fingers as features, and train them using a neural network to achieve good recognition results.
(3) In this virtual desktop interactive scene, a PRM is used to plan the robot's trajectory to avoid collisions.
(4) A model based on eye–hand cooperation with an FSM to change the robot's state is proposed. Based on the classification of previous gestures, the interaction mode is roughly divided into "instruction" and "mapping" modes. The robot deals with the fixed part based on PRM, while the human deals with the flexible part.

## 2. Experimental Setup

The system layout is shown in Figure 1. In a traditional interactive cycle, the user constantly sends out continuous commands to control the robot.

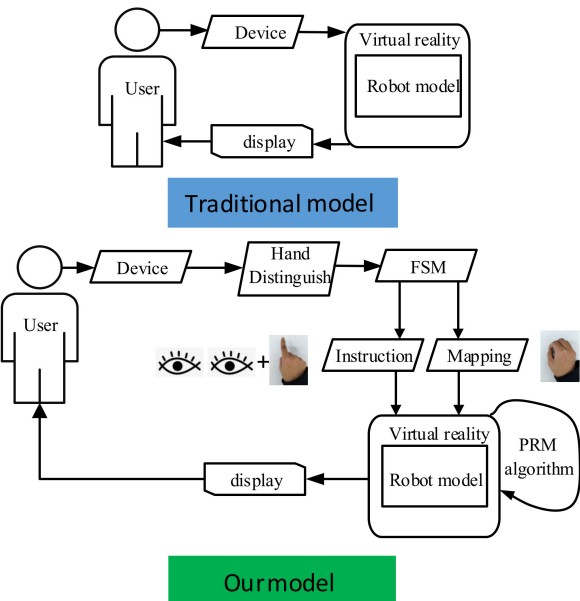

**Figure 1.** The human–robot interaction model.

In our model, the "Hand Distinguish" module extracts the position and angle information of the hand (including fingers) and then classifies gesture categories. The "FSM" is a state transfer module that inputs gestures and the current state of the robot, which are used to control the robot's operation. "Instruction" module selects objects through the eye–hand combination. "Mapping" is another interactive mode, which is entered by holding the hands tightly. Commands are divided into instructions and mapping. The instruction combined with the PRM algorithm makes the robot run automatically. The mapping enables the robot to follow the users. HRC is divided into three stages: indication, capture, and mapping. In the indication stage, users rely on hands and eyes to indicate to the robot which object should be grabbed, as well as the point where the object should be placed. During the capture stage, the robot automatically grabs the object and moves it near the specified position. After completing this stage, the mode is switched to the mapping stage by a gesture, and the hand state is mapped and controls the robot to assemble the object.

### 2.1. Components of the Test

Common methods of collecting hand movement information include control handles [29], data gloves [30], mouse-pointing, and image acquisition [31]. However, a movement towards the hand joint cannot be restored finely by control handles. Data gloves are costly and must be combined with external spatial-location equipment. Mouse-pointing is fast, but it is difficult to provide multiple degrees of freedom. This research used a non-contact Leap Motion (Leap) sensor [32], as shown in Figure 2. Leap has two cameras that can detect the space at 25–600 mm above the sensor.

The hand's space-state information can be recorded when the hand moves into this space. This study adopted Tobii Eye Tracker 4c (Tobii) [33] eye track sensor produced by Sweden to collect eye movement data, as shown in Figure 2. The range of action is 500–900 mm. The sampling rate of Leap is 120 Hz, and that of Tobii is 90 Hz. After calibration, Tobii obtains the spatial position and gaze point of the eye through a near-infrared camera. In general, Leap collects hand information while Tobii collects eye information. The robot is displayed and run in unity. The control algorithm of the robot is coded in Matlab Simulink.

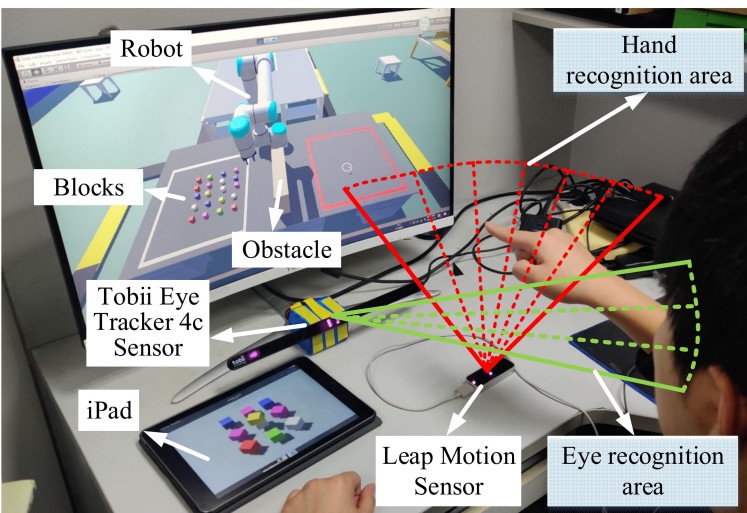

**Figure 2.** Users control the robot stacking according to the image requirements in the iPad through the eye–hand combination.

The size of the monitor used was 27 inches. During the experiment, the user faced $70 \pm 10$ cm in front of the screen, and the eye level was close to the middle of the screen. As each participant's eyes are different, such as their inter-pupillary distance, Tobii was calibrated in advance for each user according to the manufacturer's instructions. Leap was placed $30 \pm 5$ cm forward from the outside of the shoulder of the user, which is a comfortable position and did not block the Tobii equipment.

In brief, both the eye's line of sight and the finger's direction can produce an intersection on the screen. The system is designed to collect eye and hand information to improve interaction performance. Our method has two modes, namely "Instruction" and "Mapping." Here is an example of the "Instruction" mode. First, the user faces the screen and puts his hand into the Leap identification area. Then, they point to a block using the index finger (Figure 3(G1)), and their eyes also stare at the same block. The robot can automatically plan the trajectory to grasp the indicated block. When the users want to stop the grabbing action, they only need to make another gesture (Figure 3(G3)). In "Mapping" mode, the user makes a gripping gesture (Figure 3(G2)). In this mode, eye information is not needed, but the spatial position of the hand is transmitted to the robot, and the hand directly controls the robot's movement and rotation. A tablet is used to remind users to choose the block according to the shape of the picture. The reason why we combined eye–hand is that adding eye movement information can improve reliability and speed up the interactive response.

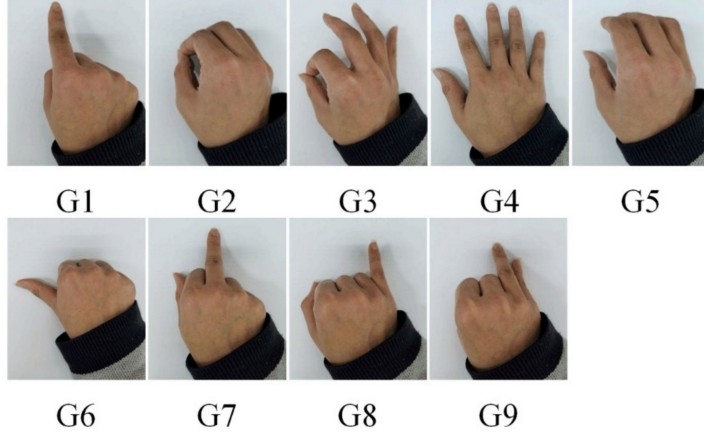

**Figure 3.** Nine gestures that are easy to use and distinguish.

## 2.2. Gesture Recognition and Object Selection

Based on previous research [34,35], the following sections list a set of typical gestures. Figure 4 shows the finger joint changes with time. In an actual process, only gestures G1–G4 are needed. The robot does not need to respond to other gestures, but considering the expansion of the function in the future, we expanded them to G1–G9. The joint angle of each gesture is maintained for 30 s, as shown in Figure 5A.

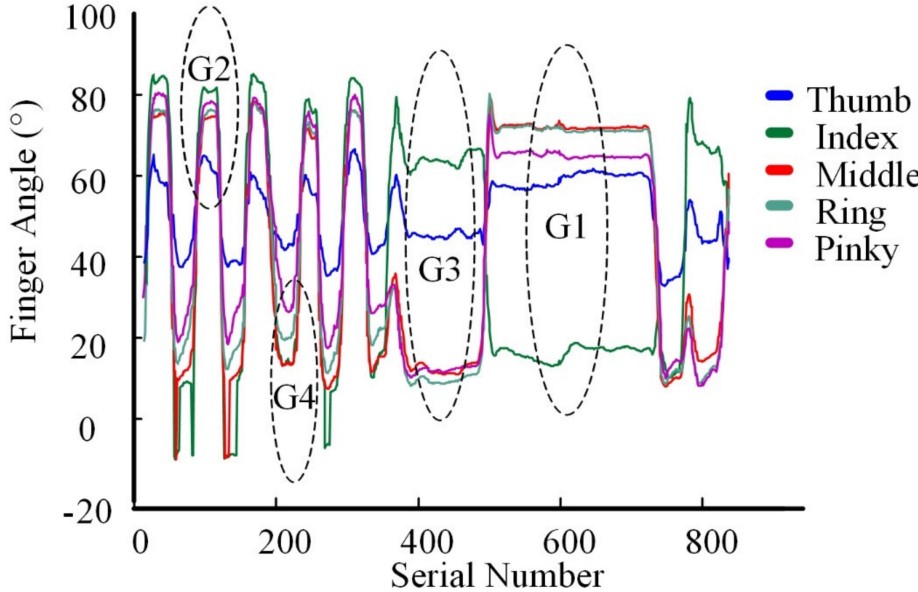

**Figure 4.** Bending angles of the finger relative to the palm.

The hidden layer from input to output. Row *H* is equal to row *W* multiplied by column *x*:

$$H_i = \sum_{i=1}^{d} W_{yi}x_i + b_i \tag{1}$$

From the hidden layer to output layer:

$$O_i = \sum_{i=1}^{d} W_{yi}x_i + b_i \tag{2}$$

Tansig is used to increase the nonlinearity:

$$T(x) = \frac{2}{1 + \exp^{-2x}} - 1 \tag{3}$$

The normalized probability is obtained by using the softmax function:

$$P(G|x) = \frac{\exp(W_G \cdot x)}{\sum_{c=1}^{C} \exp(W_c \cdot x)} \tag{4}$$

Gesture recognition [36] can be based in neural networks with two hidden layers, a Tansig activation function, and softmax classification. It can be seen that the eigenvalues of these nine kinds of algorithms are very obvious, and can recognize each gesture with 100% accuracy (Figure 5B).

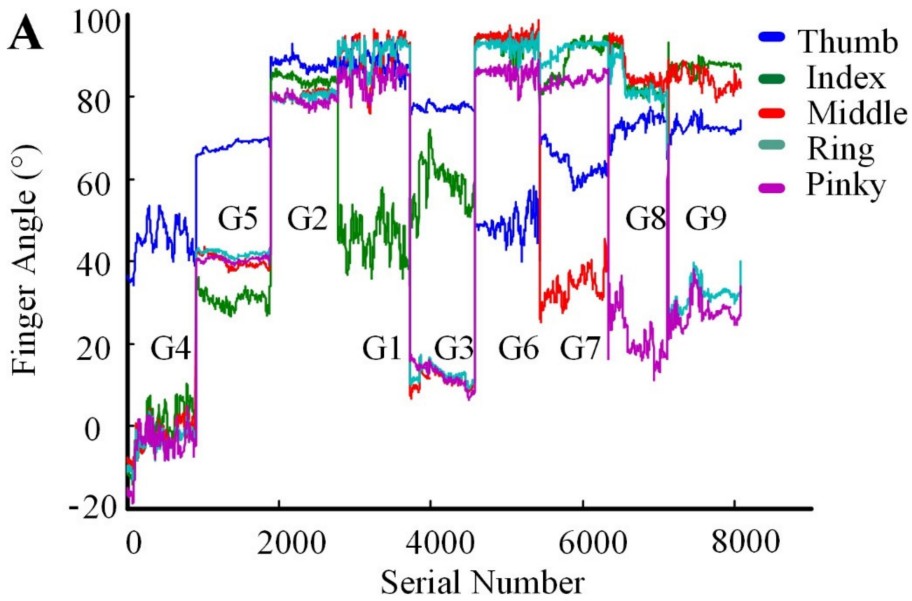

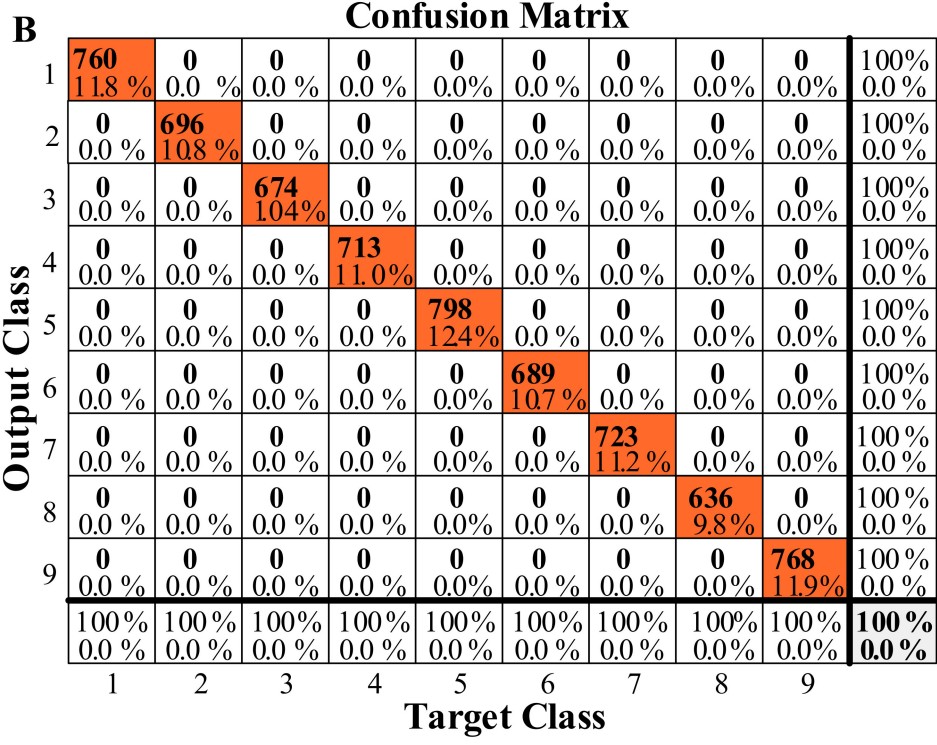

**Figure 5.** (**A**) Finger bending angles in gestures G1–G9. (**B**) The Recognition results for each gesture.

The user's focus does not always stay on one point but is changing constantly. Only useful information is used for higher-level applications of the gaze point. Five algorithms for measuring fixations and saccades, namely, I-VT, I-HMM, I-DT, I-MST, and I-AOI, were described by [37]. This paper used an algorithm similar to I-DT for object selection. The specific idea is to solve each distance between the object and the ray point. The object is selected when the eye and hand are close to the object and maintained within 200 ms of the boundary, as shown in Figure 6. Knowing the accuracy of eye and hand tracking is helpful for boundary determination. Mayer [38] removed individuals that are more than two standard deviations away from the average. For objects 1–9 (O1 to O9), we collected 3000 intersection points for each. The human eyes were 40 cm away from the computer screen. For O1, the average intersection point was 0.27 cm to the right and

0.56 cm below the target. The overall offset between the interaction and target is shown in Table 1 (comparing 40 cm and 60 cm distances to the screen).

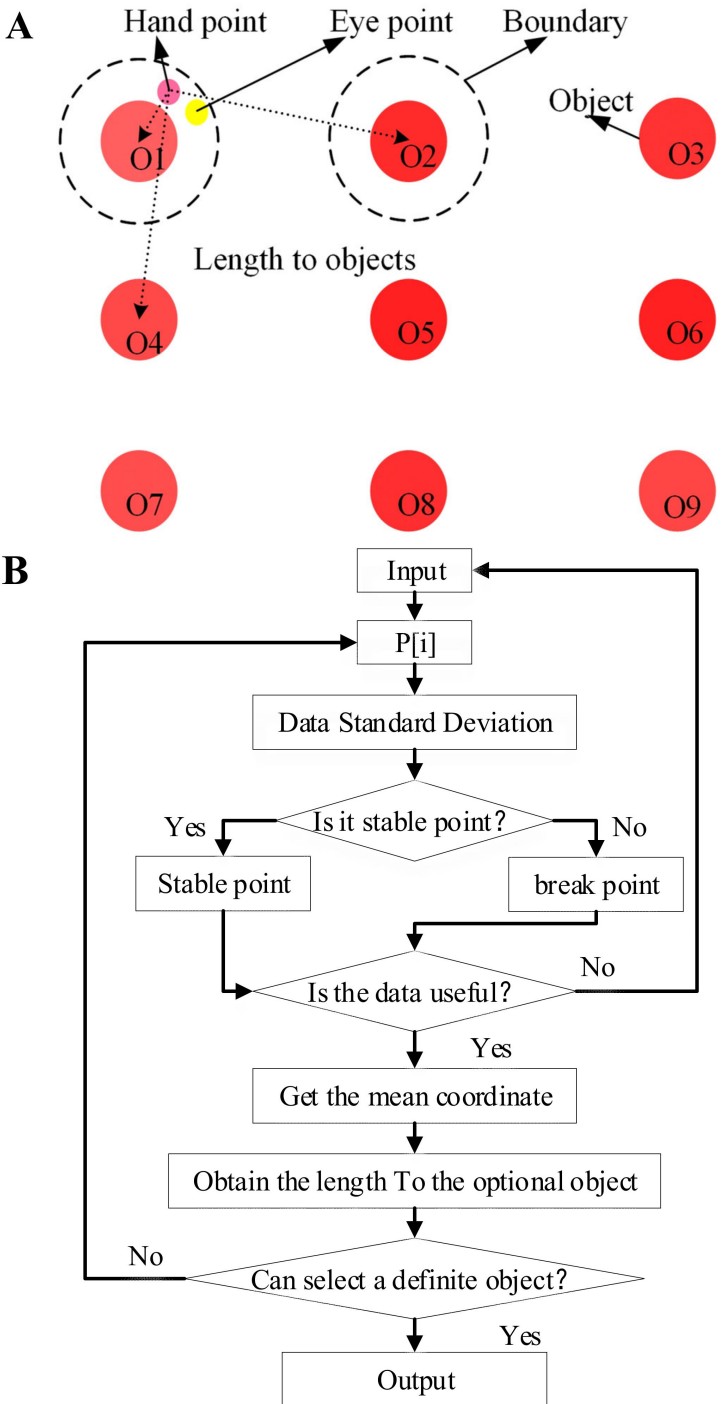

**Figure 6.** The eye and finger points used to select a target object. (**A**) Eye–hand trigger selection in the dotted line circles. (**B**) The algorithm flow chart of the judging trigger.

Similarly, we measured the jitter of the hand's indicator points (Table 2). We can see that the stability of the finger indicator points is better than that of the eye tracker. However, in experience, the stability of the hand is required to be higher. The reason is that the eye tracker is calibrated for each person and directly displays the center of the line of sight, which is equivalent to an open loop. The hand indicates points, which is only based on the subjective feeling of people. The sensor recognizes the general direction, and then the

user needs to fine-tune the finger direction according to the position of the indication point, which is equivalent to brain–hand–eye feedback. So, from experience, we hope that the indication point has less jitter. After adding a filter (Equation (5)), the variance in jitter is reduced, and the average distance from the target point is also lower.

$$\mathrm{y}_i = \alpha_i y_{i-1} + \alpha_i y_{i-2} \ldots \alpha_i y_{i-9} + \alpha_i y_{i-10} \tag{5}$$

**Table 1.** The eye track table.

| Distance | Object | Accuracy M(SD) | Object | Accuracy M(SD) | Object | Accuracy M(SD) |
|---|---|---|---|---|---|---|
| | O1 | 0.62(1.10) | O2 | 0.41(0.31) | O3 | 0.99(1.83) |
| 40 | O4 | 0.41(1.00) | O5 | 0.78(0.83) | O6 | 0.97(0.46) |
| | O7 | 0.69(1.05) | O8 | 1.26(0.58) | O9 | 0.99(0.44) |
| | O1 | 0.39(2.48) | O2 | 0.78(0.54) | O3 | 0.70(1.04) |
| 60 | O4 | 0.79(1.32) | O5 | 1.00(1.08) | O6 | 1.09(0.66) |
| | O7 | 0.63(1.89) | O8 | 0.70(2.10) | O9 | 1.00(0.91) |

Note: Distance: distance between the eye and screen, M: mean offset between the interaction and target, and SD: standard deviation. All are reported in cm.

**Table 2.** The hand track table.

| Distance | Object | Accuracy M(SD) | Object | Accuracy M(SD) | Object | Accuracy M(SD) |
|---|---|---|---|---|---|---|
| | O1 | 0.83(0.11) | O2 | 0.42(0.13) | O3 | 0.44(0.21) |
| Original | O4 | 0.98(0.32) | O5 | 0.50(0.20) | O6 | 0.87(0.46) |
| | O7 | 0.72(0.22) | O8 | 0.72(0.17) | O9 | 0.45(0.17) |
| | O1 | 0.39(0.21) | O2 | 0.65(0.23) | O3 | 0.22(0.14) |
| Filtering | O4 | 0.44(0.18) | O5 | 0.46(1.13) | O6 | 0.55(0.27) |
| | O7 | 0.37(0.26) | O8 | 0.20(0.13) | O9 | 0.56(0.12) |

### 2.3. Probabilistic Roadmap Planner

Motion planning is a basic problem in many engineering applications, such as robotics [39–42], navigation, and autonomous driving [43–45]. The essential problem in motion planning is to avoid obstacles and find a path to connect the start and target locations. Previous papers have proposed algorithms such as A* [46]. Algorithm D* [47] is an improvement on A* that supports incremental repetitive planning and has a lower calculation cost. However, the calculation cost of planning by D* is still very high and is not conducive to real-time path planning by robots. To improve the cost problem, a PRM [28] based on probability is proposed (Algorithm 1):

**Algorithm 1.** The probabilistic roadmap planner (PRM).

```
1 : while len(v_i) < npoints do
2 :   rand(v_i)
3 :   if v_i ∈ C_free
4 :     V ← V ∪ v_i
5 :     i ← i + 1
6 : end
7 : for all v_i ∈ V do
8 :   N ← the closet neighbors of v chosen from V according to disthresh
9 :   for all v_i ∈ N do
10 :    if ((v_i, v_j)) ∉ Π and LP(v_i, v_j) ≠ NIL then
11 :      Π ← Π ∪ (v_i, v_j)
12 :    end
13 : end
```

## 2.4. Finite State Machine for Human-Robot Collaboration

Switch modes are shown in Figure 7. Gesture G1 was used to indicate an object by calculating the index finger's ray position and orientation. The indication state is that the robot needs to grasp the object and put it at the target position according to G1 + E1 (gesture and eye commands). Gesture G2 was used to switch the robot mode from the indication state to the mapping state. The mapping state maps the hand space state to the end of the robot and directly commands the robot to move. Gesture G3 was used to end the mapping state. Gesture G4 was used to change the gripper to relax.

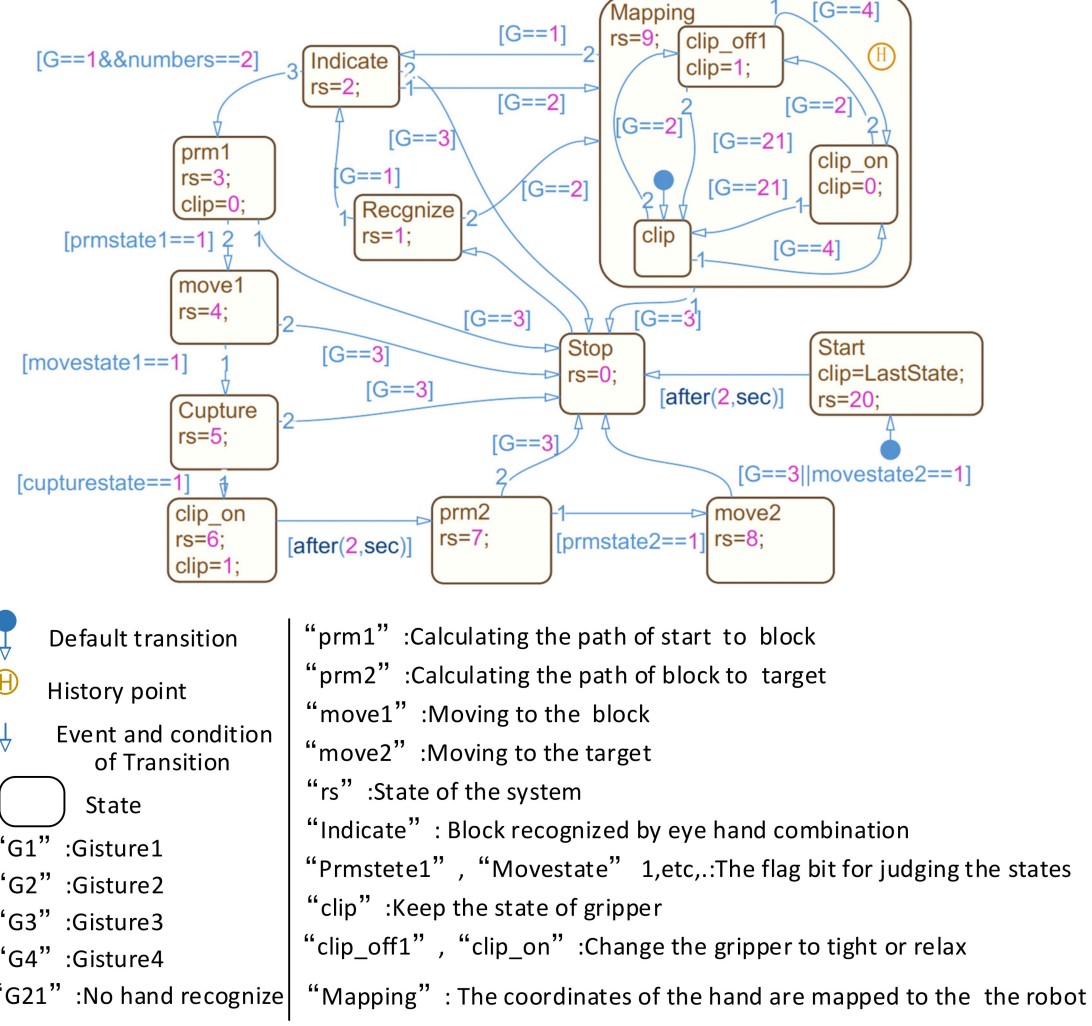

**Figure 7.** The FSM is based on Simulink. The date in the FSM is transmission to Unity by UDP connection. The user changes the running state of the robot by eye–hand combinations.

## 3. Measurement Methods

The purpose of this experiment was to determine the differences in efficiency and operation of a traditional controller method and the proposed method when controlling the robot to grasp, move, and place blocks by teleoperation in a virtual environment. The experimental environment of the participants was the same as that shown in Figure 2.

### 3.1. Experimental Scene

Building blocks reflect three major elements of a primary assembly process: acquisition, manipulation, and operation [35]. In many places, building blocks are used to describe assembly behavior. Syberfeldt [48] used building blocks in VR human–computer interaction (HCI) research. This study explored HRC by simulating the process of constructing 10

building blocks (30 mm × 30 mm × 30 mm) in cooperation with the robot of 6-DOF. The algorithm for controlling the movement of the robot gripper by hand can be seen in our previous work [49]. A model layout of the experimental scene is shown in Figure 8.

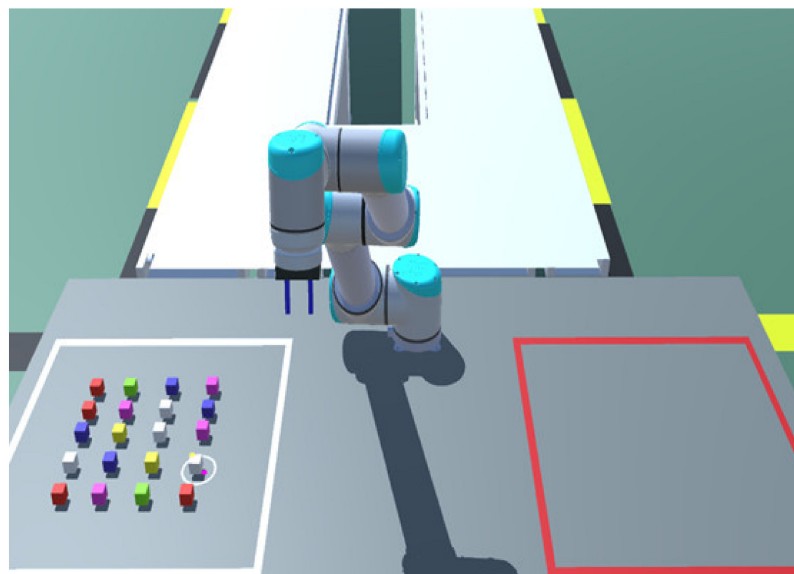

**Figure 8.** The robot selecting objects from a white square and placing them in a red square.

### 3.2. Experimental Methods

Ten volunteers (age = 22–28 years, all male) were invited to participate in the experiment without reward. The participants were all graduate students from the transmission laboratory. They were randomly assigned to two groups. Each received a five-minute explanation of how to use the device to control the robot and complete the tasks (grab blocks from the left square and place them in the right square according to the shape of a picture on a tablet). Each participant had 10 min to familiarize themselves with the operation. Group A used the traditional model, while group B used our model. To improve the reliability of the experiment, the experiment was repeated once. Before the start of the experiment, each group was introduced to each other, and the experimental operation methods and requirements were explained separately. For each round of experiments, participants were given a picture that clearly expressed the shape of the building blocks that needed to be built. All the objects could be grasped by the robot by considering their size and height. The shadows of objects were opened in experiments so that users can determine the distance between blocks and the desktop.

The flowchart for group B is shown in Figure 9. First, the users put a hand into the recognition area, and the system automatically recognized the gesture of the user, then entered the interactive state. The specific cooperation process is shown in Figure 10. According to the user's eye gaze and gesture (G1 + E1), the robot arm automatically grabbed the block and moved it to the target position. The user could change into the mapping state by using gesture G2 after the robot arm completed the movement, then map the hand state to the end effector of the robot. At this time, the user could control the movement and rotation of the end effector by the movement and rotation of their hand to complete the assembly.

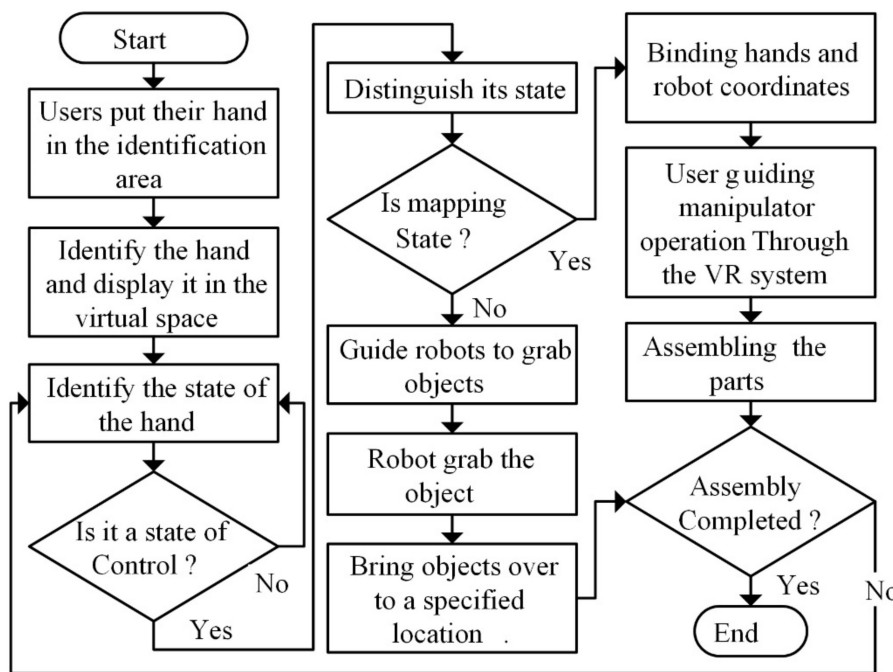

**Figure 9.** A flowchart of the human–computer interaction program.

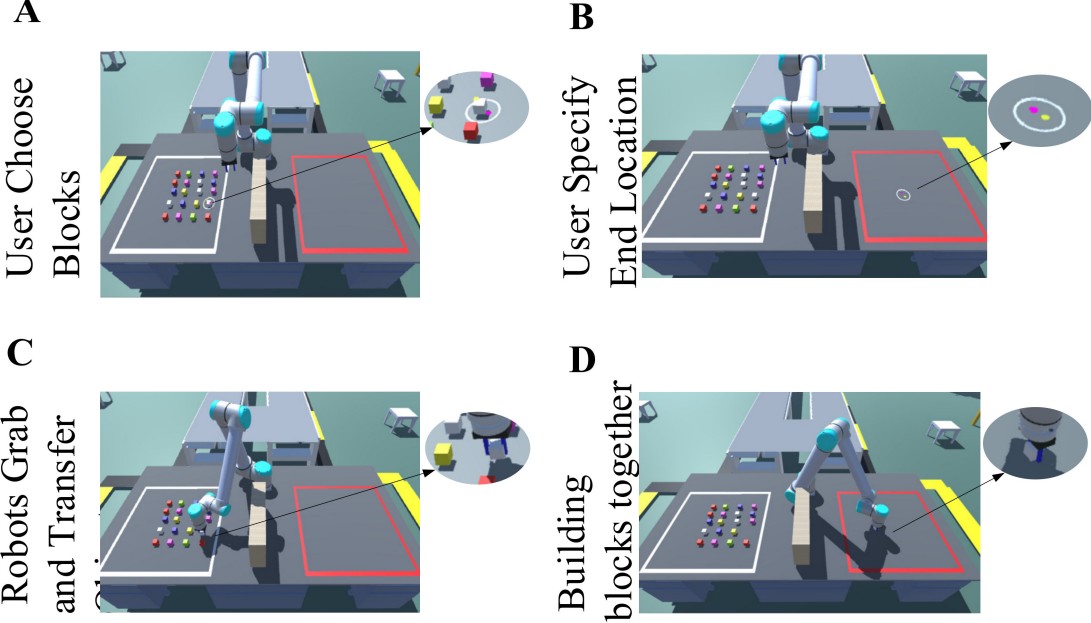

**Figure 10.** Interactive process: After the users indicate the block and position, there will be a small white circle prompt. The small red dot represents the intersection of hand and screen. The small yellow dot represents the eye annotation point.

The representative non-contact operation robot was operated through the teaching device (Figure 11). Generally, the teaching device had a touchscreen, and the movement and rotation of the robot were controlled separately. We used buttons to simulate the teaching device and control the degrees of freedom of the robot. For group, A, the user control model was replaced by a conventional robot control mode, which had six buttons (Q, W, E, R, T, Y) for controlling movements with three degrees of freedom, another six buttons (A, S, D, F, G, H) for controlling rotation with three degrees of freedom, and two more buttons (Z, X) for controlling the loosening and relaxing of the end effector. The other experimental conditions were the same as for group B.

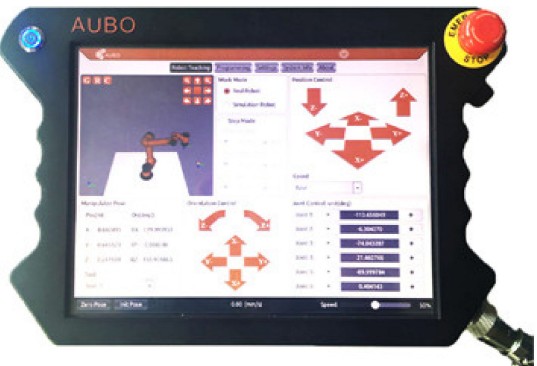

**Figure 11.** The teaching device.

## 4. Experimental Results and Discussion

The eye–hand indication trajectory result is shown in Figure 12. The green curve represents the intersection of the direction of the finger and the screen, while the blue curve represents the eye's intersection. Figure 12A shows the trajectory position on the screen, while Figure 12B shows the relationship between the trajectory and time. The selection trigger was generated at t1, which is consistent with the user's intent choice. Figure 13 is the trajectory of the robot's end movement.

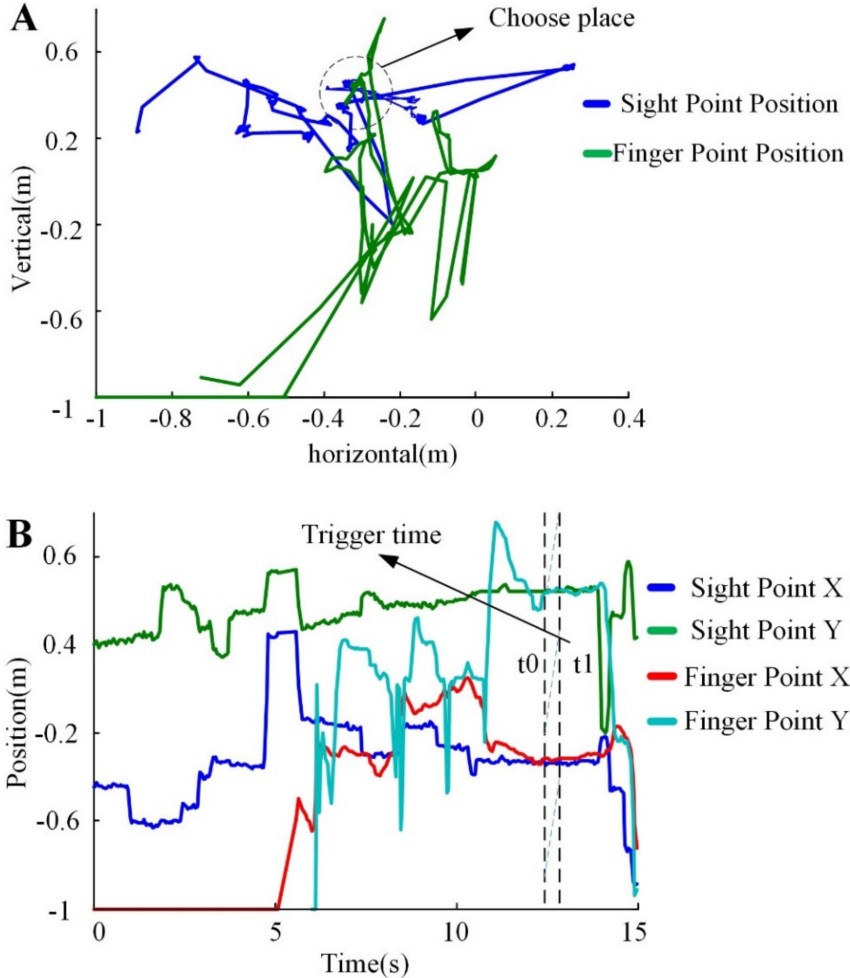

**Figure 12.** Positions of eyes and hand during the interaction. (**A**) The position of eye's point and hand's point in interaction. (**B**) The relationship between position and time of point.

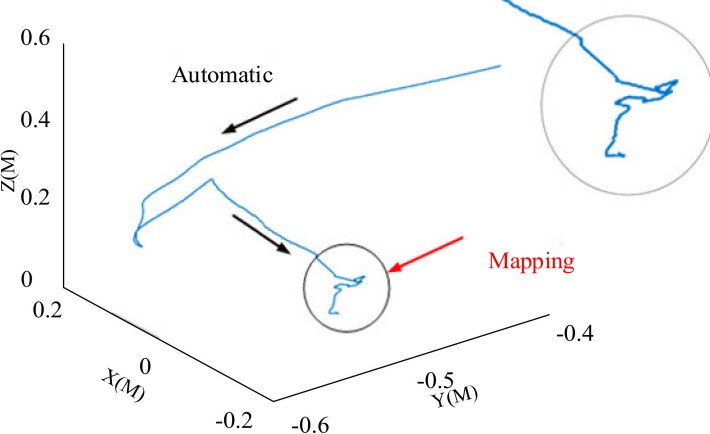

**Figure 13.** Robot trajectory of our model combining automatic and mapping parts.

## 4.1. Experimental State Change

According to the experimental process, there were 14 states in Group A (button states 1–6 = up, down, left, right, forward, and back. States 7–12 = +Roll, −Roll, +Pitch, −Pitch, +Yaw, and −Yaw. States 13–14 = Grab, Release) and four states in Group B (eye–hand states 1–4 = Recognize, Indication, Capture, and Mapping). Recording changes in the state with the time can be used to compare the performance of the experimental methods. The user participation process is presented in Figure 14. In Group A, the stage where the arm grasped and moved near to the objects took t0. The stage of grabbing the object took t1. The moving stage took t2, and the assembly stage took t3. Group B had one less stage than Group A, but the quality of the assembly should be considered in terms of the time and labor expended by users. In Group B, the stage of selecting the object and indicating the target position took t0. In the grabbing stage, the robot took t1 to capture the object and move it. Mapping assembly took t2.

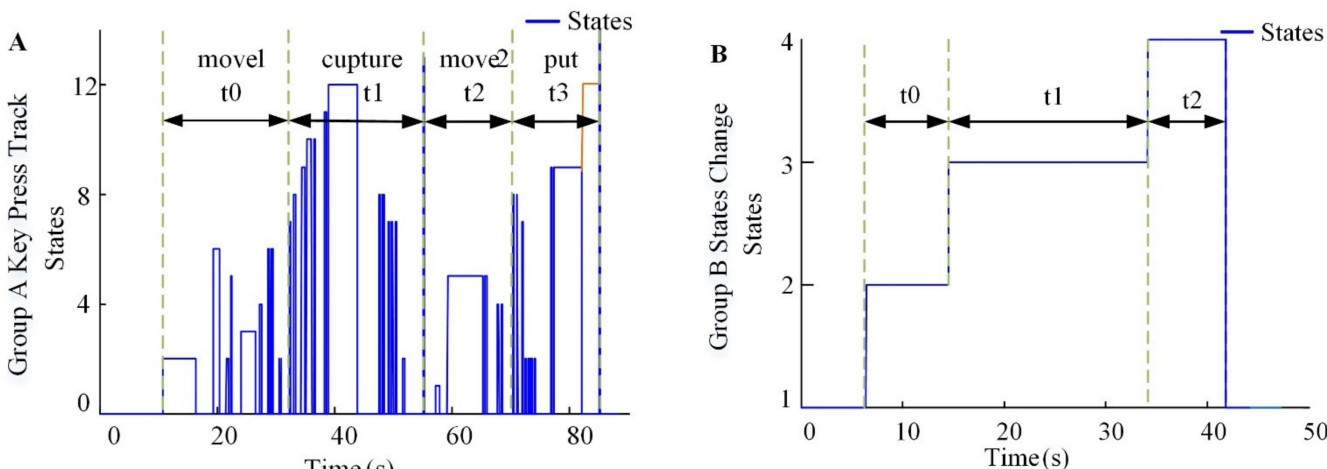

**Figure 14.** A comparison between conventional methods and the proposed method. (**A**) Button states 1–6 = up, down, left, right, forward, and back. States 7–12 = +Roll, −Roll, +Pitch, −Pitch, +Yaw, and −Yaw. States 13–14 = Grab, Release. (**B**) Eye–hand states 1–4 = Recognize, Indication, Capture, and Mapping.

## 4.2. Experimental Time Consumption

The time consumed in the experiment was the main consideration, but the percentage of time consumed in each stage of the experiment is also worth discussing. The percentage of time is mainly used to analyze the internal composition of the experimental model. It can be seen from Figure 15 that Group A completed the experiment the first time in an average of 15.2 min, while Group B took 6.2 min. For the second time, Group A completed

the experiment in an average of 13.5 min, while Group B took 5.9 min. Both times, Group A was slower than Group B. Figure 16 shows that in Group A, for the first time, 28.4% of the time was used to move the robot to the object. Grabbing objects took 22.9%, moving objects to the target position took 27.9%, and the assembly operation took 20.8%. The second time, moving the robot to the object took 29.3% of the time, grabbing objects took 22.4%, moving objects to the target position took 29.0%, and the assembly operation stage took 19.3%. For the first time for Group B, indicating the object and end position took 23.2% of the time, 55.3% was occupied by capture, moving the object to the target position, and mapping assembly of the object took 21.5%. The second time, indicating the object and end position occupied 22.0% of the time, 58.5% of the time was occupied by the capture stage, and the mapping assembly stage took 19.4%. For grasping and moving, handing objects to the robot arm significantly reduced the overall cooperative assembly time. The importance of moving the object to the target position was weakened.

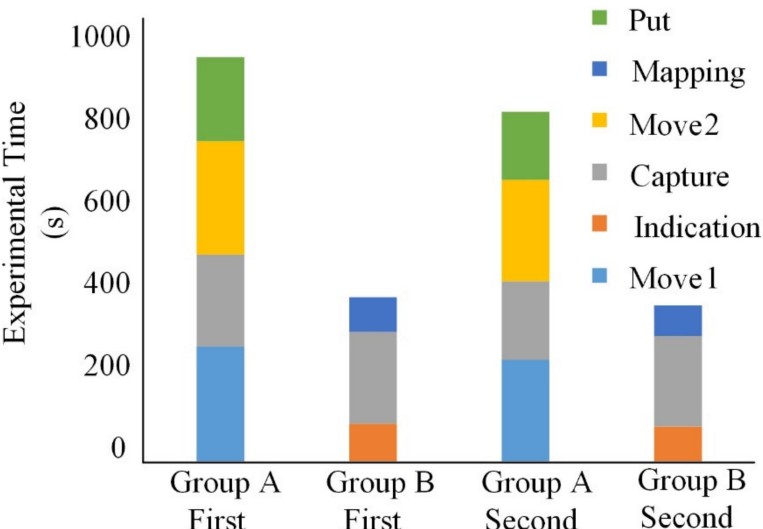

**Figure 15.** A comparison of the assembly states and times of Groups A and B.

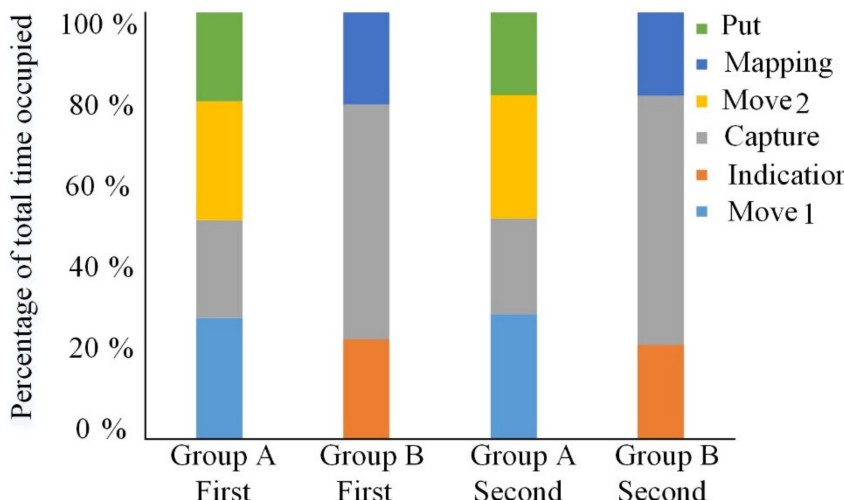

**Figure 16.** Comparisons of the percentages of time occupied by each assembly state.

It can be considered that the interaction mode combining eye and hand commands was more efficient. This may be because the traditional interaction, grab, and placement stages were all controlled by a human. Users are not adaptable to changing the 6-DOF of the robot by key. For operators will think about which degree of freedom is appropriate and whether the key is correct, which reduces the assembly efficiency. During eye–hand control,

users just watch the screen to perform manual control, a little like in Figure 13. As for the results, the first concern of users is the interaction time. The eye–hand interaction mode is more efficient than the controller interaction mode, which shows that the interaction mode of indication and mapping is feasible. Users can also change the mode of indication, such as voice or touchscreen, and compare the interaction efficiencies of the proposed and traditional controllers. Secondly, the interaction state is divided, which clearly shows the proportion of time consumed in the experiment, and readers can further optimize certain states. There have been some similar human–computer cooperation studies in which humans control robots through a data glove and the robot's complete trajectory correction and obstacle avoidance [50]. In comparison, data glove stability may be better; however, it is inconvenient to wear. Besides, collecting data from the upper body to control a robot [51]. The whole action needs to be completed by the user, and the user's burden is high. In contrast, our interaction model can be switched between instruction and mapping modes, which reduces the user burden. From the perspective of interactive experience, this paper has some shortcomings; for example, users cannot feel the grip force. Sorgini [52] controlled a robot interface via vision-based marker-less technology for hand tracking, with vibrotactile feedback delivered to the user's hand. This way of adding vibratory feedback may improve interaction adaptability to a certain extent.

## 5. Conclusions

This research aimed to solve the problems of human–robot interactive assembly. An interaction system based on eye–hand coordination with FSM in VR to control robots was proposed. Firstly, we collect hand information through Leap, select the angle of finger bending for feature learning, and use a multi-layer neural network to recognize nine kinds of gestures. The results show that all the gestures were recognized. Based on finger-pointing and eye-tracking, the block closest to the intersection of the eye–hand and screen ray is selected by the algorithm so that the robot knows what the human has focused on. Then, we used Unity to build the experimental scene. Simulink was used to control the motion and state of the robot. The robot can avoid obstacles through the probabilistic roadmap planner algorithm. Based on gestures and an FSM, the robot changes state among automatic modes, and the mapping state meets human needs. In instruction mode, the user is in a state of easy supervision and guides the robot. In mapping mode, the hand position is used to directly control the robot to complete the tasks of placement and rotation. The experiment shows that the HRC mode reduced the overall assembly time by 57.6% compared with the keypress mode, which is commonly used. The reason is that it is difficult for a user to observe the positional relationship between the block and the environment through a screen, so the user will often try pressing buttons and reduce the movement speed. Correspondingly, in the instruction state, the robot can quickly plan a path and move. This research combines human flexibility with robotic efficiency to provide better HRC performance. It may not be able to fully simulate the actual setting of a human and robot, such as delay, stability, and so on. However, it can roughly verify the algorithm of the human–computer interaction state in advance and save much time and money.

**Author Contributions:** Conceptualization, software, methodology, X.Z.; supervision, funding acquisition, X.C. and Y.H.; writing—review and editing, Z.L. All authors have read and agreed to the published version of the manuscript.

**Funding:** Chongqing Technology Innovation and application demonstration project "application demonstration research on Key Technologies of intelligent manufacturing workshop for automobile disc and sleeve parts" (No. cstc2018jszx-cyzd0697). The industrial verification platform and performance evaluation of precision machine tool spindle bearing of The National Key Research and Development Program of China (No. 2018YFB2000500). The project was supported by the Graduate Scientific Research and Innovation of Chongqing, China (Grant NO. CYB19062).

**Acknowledgments:** The authors gratefully acknowledge the support of colleagues in SKLMT of Chongqing University.

**Conflicts of Interest:** The authors declare no conflict of interest.

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
