# Peer review of "Human–Robot Collaborative Assembly Based on Eye-Hand and a Finite State Machine in a Virtual Environment"

_applsci, doi:10.3390/app11125754_

Round 1

Reviewer 1 Report

The authors propose a new assembly method. The results demostratate its efficiency compared with the traditional methods. Paper looks interesting but major revison is required:

-Introduction does not describe well the stated of the art. Please to enlarge this section and focus more in a discussion about previous methods.

-Figure 2 has very low quality

-Figure 5b is quite small. Please to enlarge it

-Figure 6b is also very small

-Figure 7 is almost imposible to read it

-Please. Enlarge Figure 11

-Conclusions are quite concise. Please to enlarge and explain much better your conclusions

Reviewer 2 Report

The authors aimed to solve the problem of human-robot interactive assembly. In the paper, they propose to use eye and hand movements as human movements.

This paper lacks a lot of information that is necessary for a thesis. I have listed below what I consider to be the missing information in this paper. However, the following items are only a small part of the missing information. In the current paper, I was not able to find the novelty or usefulness of this research.

In Section 1, the word PRM is mentioned for the first time, but there is no mention of what PRM stands for in the paper.

Figure 1 contains the words "Distinguish" and "FSM"; please use these words in your explanation of Figure 1 at the beginning of Section 2. Since I don't understand the division into instruction and mapping, please draw a diagram that corresponds to the explanatory text. Please also add an explanation of what instruction and mapping refer to.

The authors have not described how the proposed method will be used and for what purpose. Please write about the proposed method in detail, including examples. In the current text, readers cannot follow the proposed method and the experiments.
For the above, please create a section to describe the proposed method. Describe the situations in which the proposed method is intended to be used. Describe what the user does and what the robot does in the proposed method. Describe the configuration of the system. Figure 2 includes an iPad, but the reader does not understand what it is used for in the current text. Please explain why the authors chose to design the system to use the eyes and hands. Describe what tasks the eyes are used for and what tasks the hands are used for. If these are not written at the beginning of the paper, the reader will not be able to follow the story.

In Section 2.2, the following sentence is written.
"The joint angle of each gesture is maintained for some time as shown in Fig. 5 A."
Instead of writing "some time," write a specific number.

Please write the sampling rate of Leap motion and Tobii Eye Tracker 4c. Please write the official name Tobii Eye Tracker 4c when it first appears in the text as well as in the figures.

The figures should be drawn in vector. In particular, the text in Figure 6(b) is squashed and unreadable.

In the description of Section 2.1, the results of the experiment shown in Table 2 are described. However, it doesn't tell the readers how the experiment was conducted, so the readers don't know the details. The readers don't know the size of the objects or the distance between them. The readers also don't know if the participants' faces were fixed on the table or not. In order to guarantee reproducibility, the authors should describe the details to a level where the reader can do the same experiment.

Section 2.4 contains an explanation of Figure 7. However, I could not figure out what Figure 7 shows just by reading this section.

Section 3 contains a description of the experiment. However, the purpose of the experiment is not stated at the beginning of the section, and it is not clear what the experiment is intended to investigate. It is also not clear whether the environment in which the participants will conduct the experiment is the one shown in Figure 2 or not.

In Section 3.2, there is a description of the participants. However, the readers did not know what kind of people the participants were. The readers don't know if the participants are familiar with the kind of task they did in the experiment or not. Also, there is no information on how many people were divided into each group.

Figure 12 says that it is a typical eye-hand indication situation. However, I do not know what kind of data is shown. Also, I don't understand what is considered typical.

I don't understand what 14states and 4states mean in the following sentence in Section 4.1.

The results of the experiment have not been thoroughly discussed. It is not clear how the reader can make use of the results of this study.

Reviewer 3 Report

The paper deals with human-robot collaborative assembly, exploiting human tracking to control the robot during the task execution.

Comments:

  1. in the sentence "Human is responsible for controlling and monitoring production, conversely, robots do hard physical work" consider citing [1];
  2. the sentence "Various interactions, like gesture and eye movement[12, 13]." seems to have been cut;
  3. in general, a review of the state of the art for physical human-robot collaboration in assembly tasks has to be introduced ([2], [3], etc.);
  4. learning from simulation exploiting AI algorithms has to be cited in the introduction ([4], [5], etc.);
  5. the novelty of the paper is not clear. Please better define which is the novelty in your work;
  6. is it possible to include the learning of a demonstrated task in your framework? It seems that as it is, it only allows reproducing the user intention, without transferring the task knowledge to the robot;
  7. w.r.t. the proposed approach, consider citing [6] and discuss the proposed approach w.r.t. it (also in that work leap motion is used);
  8. is it possible to perform experiments with a real robot in a real assembly task?
  9. how to handle possible collisions?
  10. how the robot in simulation (or in the reality) can be controlled?
  11. carefully revise English.

[1] Vicentini, Federico, et al. "PIROS: Cooperative, safe and reconfigurable robotic companion for cnc pallets load/unload stations." Bringing Innovative Robotic Technologies from Research Labs to Industrial End-users. Springer, Cham, 2020. 57-96.

[2] Roveda, Loris, et al. "Human–robot collaboration in sensorless assembly task learning enhanced by uncertainties adaptation via Bayesian Optimization." Robotics and Autonomous Systems 136 (2021): 103711.

[3] Akkaladevi, Sharath Chandra, et al. "Programming by Interactive Demonstration for a Human Robot Collaborative Assembly." Procedia Manufacturing 51 (2020): 148-155.

[4] Shahid, Asad Ali, et al. "Learning Continuous Control Actions for Robotic Grasping with Reinforcement Learning." 2020 IEEE International Conference on Systems, Man, and Cybernetics (SMC). IEEE, 2020.

[5] Apolinarska, Aleksandra Anna, et al. "Robotic assembly of timber joints using reinforcement learning." Automation in Construction 125 (2021): 103569.

[6] Sorgini, Francesca, et al. "Tactile sensing with gesture-controlled collaborative robot." 2020 IEEE International Workshop on Metrology for Industry 4.0 & IoT. IEEE, 2020.

Round 2

Reviewer 1 Report

Paper is now acceptable for publication

Author Response

Thank you for your review.

Reviewer 2 Report

The reviewers confirmed that the authors responded appropriately to the comments.

Author Response

Thank you for your review.

Reviewer 3 Report

The reviewer cannot see the reply to the provided comments. Please upload it.

Author Response

Sorry for miss loaded. Please see the attachment.

Round 3

Reviewer 3 Report

The paper can now be accepted.